# Ex-VAD: Explainable Fine-grained Video Anomaly Detection Based on Visual-Language Models

Chao Huang[1]  Yushu Shi[1]  Jie Wen[2]  Wei Wang[1]  Yong Xu[2]  Xiaochun Cao[1] *

## Abstract

With advancements in visual language models (VLMs) and large language models (LLMs), video anomaly detection (VAD) has progressed beyond binary classification to fine-grained categorization and multidimensional analysis. However, existing methods focus mainly on coarse-grained detection, lacking anomaly explanations. To address these challenges, we propose **Ex-VAD**, an **Ex**plainable Fine-grained **V**ideo **A**nomaly **D**etection approach that combines fine-grained classification with detailed explanations of anomalies. First, we use a VLM to extract frame-level captions, and an LLM converts them to video-level explanations, enhancing the model's explainability. Second, integrating textual explanations of anomalies with visual information greatly enhances the model's anomaly detection capability. Finally, we apply label-enhanced alignment to optimize feature fusion, enabling precise fine-grained detection. Extensive experimental results on the UCF-Crime and XD-Violence datasets demonstrate that **Ex-VAD** significantly outperforms existing State-of-The-Art methods.

## 1. Introduction

Video Anomaly Detection (VAD) is an important technology with a wide range of applications that cover areas such as security surveillance, healthcare, autonomous driving, and content auditing (Zhao et al., 2017; Wang et al., 2019; Samaila et al., 2024). It aims to improve the safety and efficiency of systems by automatically identifying anomalous events or behaviors through the analysis of video data (Ren

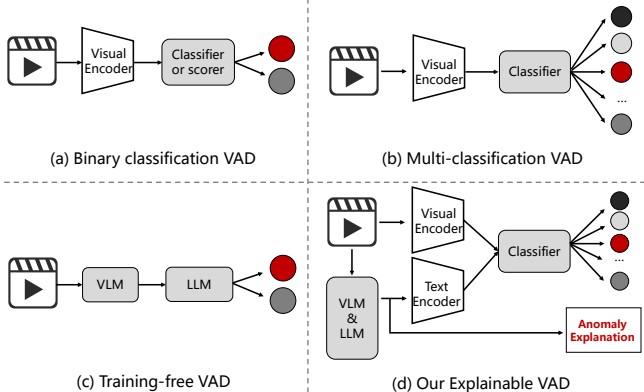

Figure 1. Recent research in VAD can be categorized into three types: a) Traditional binary classification VAD, b) Multi-classification VAD, and c) Training-free VAD. Building on the optimization of these approaches, our Ex-VAD is presented as: d) Explainable VAD based on VLMs and LLMs.

et al., 2021; Nawaratne et al., 2020). For example, rapid detection of dangerous behaviors for timely intervention in surveillance, detection of abnormal road conditions to avoid accidents in autonomous driving, and identification of abnormal vital signs to provide timely assistance in healthcare monitoring.

Traditional VADs (Huang et al., 2024; Wang et al., 2025; Huang et al., 2023; 2022; Ramachandra et al., 2022; Liu et al., 2024; Zaigham Zaheer et al., 2020; Yan et al., 2023) typically coarse-grained analyze videos, determining only whether a video contains abnormal behavior and categorizing it as normal or anomalous. However, such approaches (Nguyen & Meunier, 2019) face significant limitations in practical applications. First, coarse-grained detection fails to provide detailed descriptions of specific types of abnormal behavior, which is inadequate in scenarios that require tailored responses to distinct anomalies, for example, addressing varying security threats in surveillance systems or diagnosing multiple abnormal conditions in medical monitoring. Second, coarse-grained methods are easily influenced by the complexity of video backgrounds and the diversity of scenes, making it challenging to pinpoint the time and type of anomalies accurately. This deficiency com-

[1]Shenzhen Campus of Sun Yat-Sen University, School of Cyber Science and Technology, Shenzhen, China [2]Harbin Institute of Technology, School of Computer Science and Technology, Shenzhen, China. Correspondence to: Xiaochun Cao <caoxiaochun@mail.sysu.edu.cn>.

*Proceedings of the 42$^{st}$ International Conference on Machine Learning*, Vancouver, Canada. PMLR 267, 2025. Copyright 2025 by the author(s).

promises both the detection accuracy of the system and its response efficiency.

Fine-grained video anomaly detection becomes particularly important to distinguish different types of anomalous behavior further and provide more targeted and interpretative detection results. In recent years, visual-language pre-training (VLP) models such as CLIP (Radford et al., 2021) have significantly improved the semantic representation of images and text through contrast learning, driving advances in visual representation. CLIP-based task-specific models have excelled in various visual tasks, achieving unprecedented performance breakthroughs. In VAD, some researchers (Wu et al., 2024c;a) have used CLIP's image-text alignment to achieve fine-grained anomaly detection.

Despite advances, existing methods still struggle to explain anomalous behavior effectively. Even when anomalous events are successfully detected, models often fail to provide clear explanations for the causes of the anomalies, posing significant challenges to decision-makers. For example, in security monitoring, the detection of abnormal behavior in a specific area without a clear explanation can complicate subsequent response efforts, leading to inefficiency and delays. Consequently, enhancing the interpretability of VAD has become a crucial focus in the field's development. Recently, the rapid progress in LLMs has introduced new possibilities for VAD. Some researchers (Zanella et al., 2024; Ye et al., 2024) have proposed training-free anomaly detection methods by generating descriptive text explanations of anomalies using VLMs and LLMs. However, these methods primarily rely on the generated text for anomaly detection, often neglecting the full potential of the visual modality. Other researchers (Lv & Sun, 2024; Kim et al., 2023a; Tang et al., 2024b) have achieved interpretable anomaly detection by fine-tuning large models. While effective, these approaches often result in complex models that may be challenging to deploy and maintain.

To address these challenges, we propose a novel method called Ex-VAD, which is designed to overcome the limitations of traditional VAD methods, particularly in fine-grained classification and anomaly explanation. Specifically, we first propose an Anomaly Explanation Generation Module (AEGM), which extracts frame-level captions from videos using VLMs, followed by a cleaning step to refine the captions. The cleaned captions are then integrated by an LLM to generate video-level anomaly explanations through specific prompts, which enable the model to detect abnormal behavior in the video and analyze its cause. Second, we develop a Multimodal Anomaly Detection Module (MADM), which encodes the text from AEGM and extracts both temporal and spatial features between video frames. These features are then fed into a coarse-grained anomaly classifier to determine whether the video contains anoma-

lies. Finally, we employ a Label Augment and Alignment Module (LAAM), which uses an LLM to expand anomaly category labels into phrases, selects the top-$k$ phrases semantically most similar to the original labels, and aligns them with the fused multimodal features to obtain fine-grained anomaly categories. In summary, Ex-VAD effectively integrates multimodal features, fine-grained classification, and anomaly explanations, providing a comprehensive solution to video anomaly detection with enhanced interpretability and accuracy.

Our main contributions are summarized as follows.

- We develop an Anomaly Explanation Generation Module (AEGM), which utilizes a VLM and an LLM to generate explanations for video anomalies, allowing the model to detect abnormal behavior and analyze its cause, thereby enhancing its semantic interpretation.

- We propose a Label Augment and Alignment Module (LAAM) that enhances label semantics, enabling the model to better align videos with anomaly categories, thereby improving fine-grained anomaly classification, particularly for complex categories.

- Extensive experimental results show that our method outperforms existing approaches in both coarse-grained and fine-grained accuracy, improving overall anomaly detection and classification precision.

## 2. Related Work

### 2.1. Video Anomaly Detection

According to the output of existing VAD, it can be divided into binary-classification VAD (Ramachandra et al., 2022; Liu et al., 2024), multi-classification VAD (Sultani et al., 2019; Wu et al., 2024a;c), and interpretable VAD (Lv & Sun, 2024). Traditional VAD methods classify videos as normal or abnormal. They typically adopt a classification paradigm. Firstly, pre-trained visual models are used to extract frame-level features. Then, these features are fed into a binary classifier based on Multiple Instance Learning (MIL) for training. Finally, abnormal events are detected based on the predicted anomaly confidences.

With the development of the CLIP model, some methods have attempted to make improvements. VadCLIP (Wu et al., 2024c) proposed a fine-grained Weakly Supervised Video Anomaly Detection (WSVAD) method that can distinguish different types of abnormal frames. VadCLIP encodes text labels into class embeddings and calculates the matching similarities between class embeddings and frame-level visual features to obtain an alignment map. Each input text label represents a class of abnormal events, thus achieving fine-grained WSVAD.

Interpretability is of utmost importance in VAD, especially in sensitive or high-stake applications. Early methods often relied on black-box models, and their prediction results were difficult to trust. Recently, some methods have utilized Large Language Models (LLMs) and Vision-Language Models (VLMs) to generate understandable reasoning through semantic insights and textual explanations. For example, VADor (Lv & Sun, 2024) fine-tunes the projection layer of VideoLLaMA to integrate anomaly detection with semantic reasoning. HAWK (Tang et al., 2024a)enhances interpretability by integrating motion-based reasoning through interactive VLMs. However, there are still challenges in balancing the granularity of explanations and computational efficiency.

### 2.2. Visual Language Model in VAD

Vision language models (VLMs) offer a new perspective for detecting anomalies in video anomaly detection (VAD), especially in fine-grained classification and explanation of anomalous behaviors. Traditional VAD methods (Tian et al., 2021a; Li et al., 2022b;a) mainly focus on identifying anomalous behaviors in videos but lack detailed classification of these behaviors. (Wu et al., 2024c) leverages the pre-trained CLIP model to align video frames with labels in VAD, enabling fine-grained anomaly classification. Meanwhile, the use of LLMs in VAD is still in its infancy(Kim et al., 2023b) and LAVAD (Zanella et al., 2024) implemented training-free VAD using pre-trained LLMs and VLMs. This method efficiently transforms LLMs into video anomaly detectors by generating textual descriptions of each frame in the test video, which is combined with prompting to activate LLMs for time series aggregation and anomaly score estimation. Additionally, by referring to VLMs, we establish a strong complementary relationship between visual and textual modalities. This approach not only enables the detection of anomalous behaviors but also provides clear explanations for each behavior, enhancing the explanation of anomaly detection.

### 2.3. Prompt Learning

Prompt learning, a technique for adapting prompt words to fit a specific task, was initially applied mainly in the field of Natural Language Processing (NLP) and has gradually been extended to the visual domain. CLIP (Radford et al., 2021) relys on fixed hand-designed cues (e.g., a photo of a class), which are suitable for open domains but not flexible enough. CLIP-COOP (Zhou et al., 2022a) introduces learnable context vectors, enhancing performance with limited samples but struggling with generalization. These advances refine prompt adaptation, improving vision-language models across diverse tasks. In VAD, VADCLIP leverages trainable textual templates to generate precise anomaly descriptions. However, manually designing prompts remains

time-consuming and highly sensitive to template content. To address this challenge and reduce the dependence on hand-crafted language designs, PEL4VAD (Pu et al., 2024) used ConceptNet definitions to create prompt templates and expanded class labels through a conceptual dictionary, significantly improving open-vocabulary object detection. Based on this approach, this paper uses GPT4 (OpenAI & etc, 2024) to generate rich semantics for simple labels, and uses CLIP image-text alignment to allow the VAD model to achieve better performance in fine-grained anomaly classification.

## 3. Approach

### 3.1. Architecture

As shown in Figure 2, the proposed Ex-VAD consists of three components: an Anomaly Explanation Generation Module (AEGM), a Multimodal Anomaly Detection Module (MADM), and a Label Augment and Alignment Module (LAAM). Ex-VAD processes input videos **V** by first utilizing the AEGM to generate anomaly explanation text **E**. This text serves two purposes: providing interpretative explanations for video anomalies and acting as the text modality input for the MADM, where it is fused with visual features for coarse-grained anomaly detection. Finally, the LAAM refines the detection by expanding and aligning labels to achieve fine-grained anomaly classification, ensuring both interpretability and accuracy in video anomaly detection. The implementation details are introduced as follows.

### 3.2. Anomaly Explanation Generation Module

LAVAD (Zanella et al., 2024) demonstrated the feasibility of achieving anomaly detection by prompting VLMs and LLMs to generate text descriptions. Inspired by this approach, our AEGM improves the prompting mechanism to guide LLMs in time series aggregation and the generation of anomaly explanations. This not only helps the visual module enhance the performance of VAD but also serves as an explanation for the causes of anomalies, further enhancing the interpretability of detection. As shown in Figure 3, AEGM consists of two sub-modules: the Caption Extraction and Cleaning Module, and the Explainable Modules Based on LLM.

**Caption Extraction and Cleaning Module.** With the rapid development of VLMs, the ability to generate captions from videos has become increasingly powerful. First, uniformly sample $n$ frames from the video $V$. For each frame $I_i \in V$, we use the SOTA captioning model $\Phi_C$ i.e. BLIP-2 (Li et al., 2023) and set appropriate prompts $P_C$ to generate frame-level text descriptions:

$$T_i = P_C \cdot \Phi C(I_i). \tag{1}$$

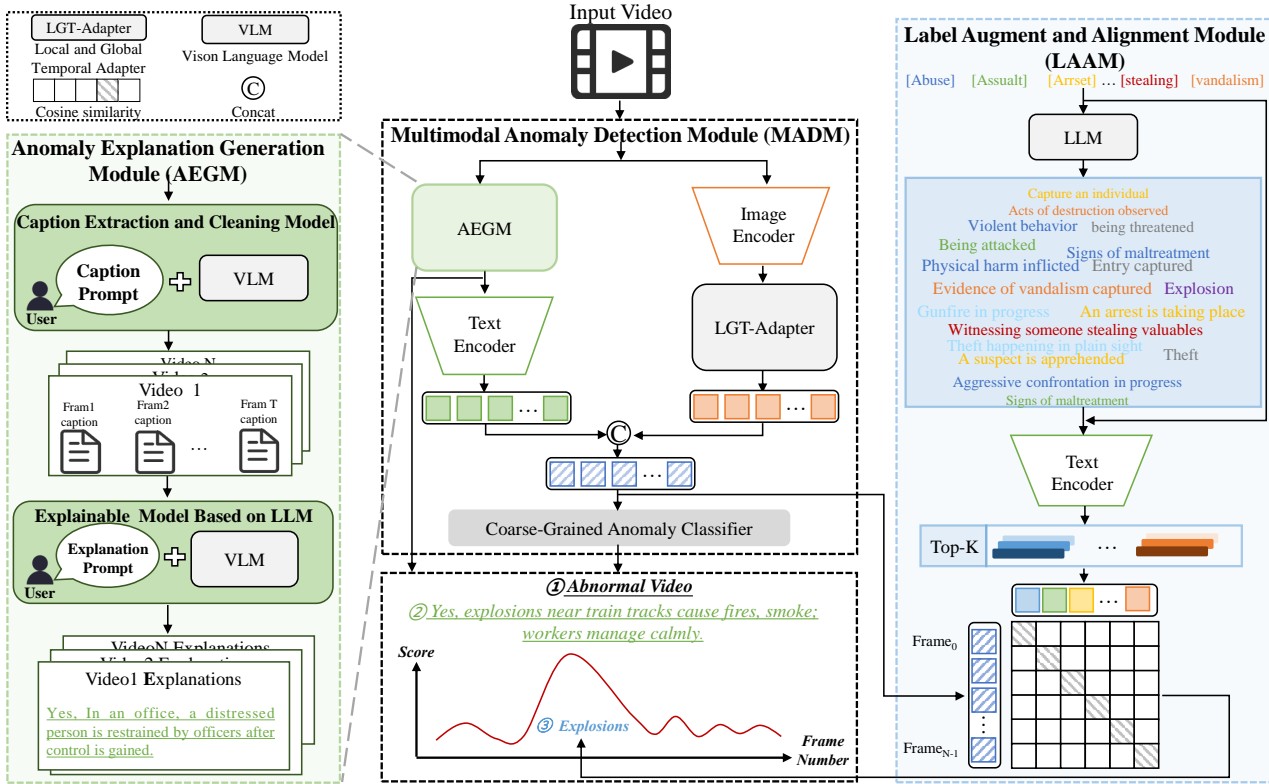

*Figure 2.* Our Ex-VAD includes three components: an Anomaly Explanation Generation Module using VLM and LLM to generate anomaly explanation text, a Multimodal Anomaly Detection Module combining enriched visual and textual features for coarse anomaly classification, and a Label Augment and Alignment Module that refines the detection by expanding and aligning labels to achieve fine-grained anomaly classification.

Due to the randomness of VLMs, some irrelevant captions may be generated, which may harm training. Since the scenes in the video are captured by a static camera at a high frame rate, the semantic content between frames overlaps to some extent. From this perspective, we alleviate the above problems by designing an image-text alignment mechanism. Specifically, we use a vision-language encoder to encode the captions of each frame. For each frame $I_i \in V$, we calculate its closest caption:

$$\hat{T}_i = \arg\max_{T \in \mathcal{T}} \{E_I(\mathbb{I}_i) \cdot E_T(T)\}, \quad (2)$$

where $\{\cdot, \cdot\}$ is the cosine similarity, $E_I$ is the image encoder of the VLM, $E_T$ is the text encoder and $T = \{T_1, ..., T_N\}$. This module allows us to generate fairly accurate text descriptions for each video frame.

**Explainable Modules Based on LLM.** The cleaned captions can describe frame information more accurately than the initial captions, but they are only simple descriptions and cannot describe abnormal phenomena in detail. Therefore, we prompt LLM i.e. LLAMA-3 (Touvron et al., 2023)to generate the required anomaly explanations. Specifically, we input the collection $\hat{T}$ of cleaned frame captions and

the prompt $P_S$ into the advanced LLM $\Phi_{LLM}$ to obtain the explanation $E$ for video $V$:

$$E = P_S \cdot \Phi_{LLM}(\hat{T}), \quad (3)$$

where $\hat{T} = \{\hat{T}_1, \hat{T}_2, ..., \hat{T}_N\}$. Through the above methods, we can obtain an anomaly description $E$ that is more accurate semantically and temporally than the captions $\hat{T}$.

### 3.3. Multi-Modal Feature Fusion

This component primarily performs coarse-grained anomaly detection by entering the fused visual and text features into an anomaly classifier. For visual features, we follow prior work (Wu et al., 2024c) to uniformly sample dense video frames from the input video at 16-frame intervals, obtaining a video frame sequence $V$. The video frames are then encoded by the frozen visual encoder $E_I$ in CLIP to produce frame features $F_I$. To bridge the gap between the image and video domains in CLIP, we adopt the approach from (Wu et al., 2024c), modeling the temporal dependencies of the video frame sequence using the Local and Global Temporal Adapter (LGT-Adapter):

$$F_V = LGT(E_I(V)). \quad (4)$$

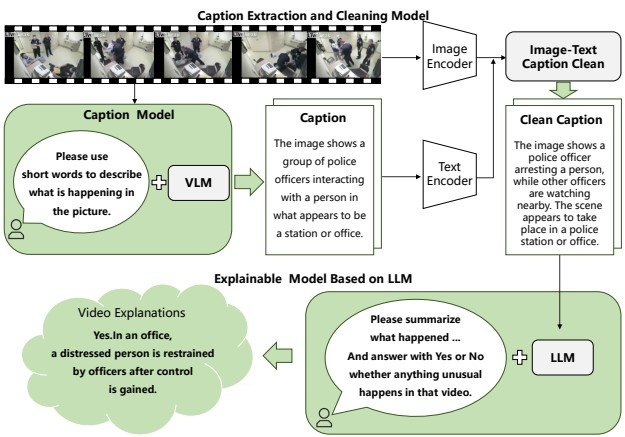

*Figure 3.* The Anomaly Explanation model first generates video frame captions via VLM, then cleans up the frame captions using the image-text module, and finally generates a detailed video interpretation using LLM. This interpretation is later used as a textual modality to enhance the performance of anomaly detection in conjunction with the visual modality, in addition to being used as an anomaly interpretation for the video.

We use the exception of explanation $E$ generated from AEGM as textual information. These textual information are encoded by the frozen textual encoder $E_T$ in CLIP to produce textual features $F_T = E_T(E)$. Subsequently, the textual features and visual features are fused into $F_F = F_V + F_T$, which is then input into a binary classifier that contains a feed-forward network (FFN) layer, an FC layer, and a Sigmoid activation to obtain the anomaly scores $s \in R^{n*1}$:

$$s = \text{Sigmoid}(FC(FFN(F_F) + F_F)). \quad (5)$$

### 3.4. Label Augment and Alignment Module

This part mainly includes the following two steps: label augmentation set construction and fine-grained classification.

**Label Augmentation Set Construction.** We utilize a pre-trained LLM (OpenAI & etc, 2024) to generate $m$ descriptive sentences related to each category label. To filter the sentences that are most semantically related to the category labels, we calculate the semantic similarity between the category labels and the generated sentences by using cosine similarity. Specifically, first, the category labels $L$ and the related descriptive sentences $\{S_1, ..., S_m\}$ generated by the LLM are encoded into vectors. Then, the cosine similarity between the label vector and the sentence vector is calculated as:

$$\text{Sim}(L, S_i) = \frac{\mathbf{v}_L \cdot \mathbf{v}_{S_i}}{\|\mathbf{v}_L\| \cdot \|\mathbf{v}_{S_i}\|}, \quad (6)$$

where $\mathbf{v}_L$ and $\mathbf{v}_{S_i}$ represent the embedding vectors of the category label and the generated sentence, respectively. Ac-

cording to the similarity score, the top-k sentences with the highest similarity are selected from the generated sentences. The features of the screened sentences are integrated with the original label embeddings to form the final enhanced label embeddings $F_L$.

**Fine-grained Classification.** Calculate the matching similarity between these category embeddings $F_L$ and Fusion features $F_F$ to obtain an alignment map $M \in R^{n*m}$, where $m$ is the number of text labels. In this alignment map, each input text label represents a class of abnormal events. By analyzing the similarity between the video and different category labels, a more detailed classification of abnormal events is achieved, naturally achieving the goal of fine-grained classification.

### 3.5. Loss Function

**Binary Classification Loss.** We follow previous work (Wu et al., 2020) and use the Top-k mechanism to select $K$ highest anomaly confidence levels among anomalous and normal videos as video-level predictions. The classification loss $\mathcal{L}_{BCE}$ is then computed using the binary cross-entropy between video-level prediction and ground truth:

$$\mathcal{L}_{BCE} = -[y \log(s) + (1 - y) \log(1 - s)], \quad (7)$$

where $s$ denotes the predicted score and $y$ is the true label (usually 0 represents normal, and 1 represents abnormal).

**Multiple Class Loss.** For multi-classification tasks, we propose the MIL-Align mechanism to align the frame-level fusion feature $F_F$ and all label embeddings $F_L$. Specifically, for each video, we select the top-k similarity values and compute the average to measure how well this video is aligned with the current class. Then, we can obtain a vector $V = \{v_1, \ldots, v_m\}$ that represents the similarity between this video and all classes. We hope the video and its paired textual label emit the highest similarity score among others. To achieve this, the multi-class prediction is first computed as follows:

$$p_i = \frac{\exp(v_i/\tau)}{\sum_j \exp(v_j/\tau)}, \quad (8)$$

where $p_i$ is the prediction with respect to the $i$th class, and $\tau$ refers to the temperature hyper-parameter for scaling. Finally, the alignment loss $\mathcal{L}_{MCE}$ can be computed by the cross-entropy:

$$\mathcal{L}_{MCE} = -\sum_{i=1}^{m} y_i \log(p_i), \quad (9)$$

where $y_i$ is the ground truth label and $m$ is the total number of classes.

**Contrastive Loss.** To pull apart the normal class embeddings from the anomaly class embeddings, we introduce

| Method | mAP@IOU(%) | | | | | |
|---|---|---|---|---|---|---|
| | 0.1 | 0.2 | 0.3 | 0.4 | 0.5 | AVG |
| Random Baseline | 0.21 | 0.14 | 0.04 | 0.02 | 0.01 | 0.08 |
| RealAD (2018) | 5.73 | 4.41 | 2.69 | 1.93 | 1.44 | 3.24 |
| RTFM (2021) | 12.59 | 7.54 | 6.44 | 5.42 | 1.54 | 6.71 |
| AVVD (2022) | 10.27 | 7.01 | 6.25 | 3.42 | 3.29 | 6.05 |
| DMU(2023) | 11.32 | 7.62 | 5.97 | 4.33 | 2.36 | 6.32 |
| CLIP-TSA(2023) | 12.62 | 8.13 | 6.66 | 4.28 | 1.91 | 6.72 |
| UMIL(2024) | 11.84 | 7.85 | 6.52 | 3.97 | 2.84 | 6.60 |
| VadCLIP(2024) | 11.72 | 7.83 | 6.40 | 4.53 | 2.93 | 6.68 |
| STPrompt(2024) | 11.56 | 7.49 | 6.13 | 5.11 | 2.11 | 6.48 |
| **Ex-VAD (Ours)** | **16.51** | **12.35** | **9.41** | **7.82** | **4.65** | **10.15** |

*Table 1.* Fine-grained comparisons on UCF-Crime.

| Method | mAP@IOU(%) | | | | | |
|---|---|---|---|---|---|---|
| | 0.1 | 0.2 | 0.3 | 0.4 | 0.5 | AVG |
| Random Baseline | 1.82 | 0.92 | 0.48 | 0.23 | 0.09 | 0.71 |
| RealAD (2018) | 22.72 | 15.57 | 9.98 | 6.20 | 3.78 | 11.65 |
| RTFM (2021) | 31.25 | 26.85 | 21.94 | 13.56 | 12.54 | 21.23 |
| AVVD (2022) | 30.51 | 25.75 | 20.18 | 14.83 | 9.79 | 20.21 |
| DMU(2023) | 32.33 | 28.88 | 22.57 | 14.33 | 13.68 | 22.36 |
| CLIP-TSA(2023) | 34.53 | 32.88 | 28.11 | 13.65 | 10.01 | 23.84 |
| UMIL(2024) | 34.44 | 27.13 | 22.63 | 19.85 | 13.24 | 23.46 |
| VadCLIP(2024) | 37.03 | 30.84 | 23.38 | 17.09 | 14.31 | 24.70 |
| STPrompt(2024) | 38.21 | 25.63 | 28.66 | 13.11 | 11.63 | 23.44 |
| **Ex-VAD (Ours)** | **40.14** | **32.75** | **28.78** | **20.15** | **18.35** | **28.23** |

*Table 2.* Fine-grained comparisons on XD-Violence.

the contrast loss. Specifically, we first calculate the cosine similarity between the normal class embedding and other abnormal class embeddings, and then compute the contrastive loss $\mathcal{L}_{cts}$ as follows:

$$\mathcal{L}_{CTS} = \sum_j \max\left(0, \frac{L_N^T L_{A_j}}{\|L_N\|_2 \cdot \|L_{A_j}\|_2}\right), \qquad (10)$$

where $L_N$ is the normal class embedding, and $L_{A_j}$ is the abnormal class embedding.

Overall, the final total objective of Ex-VAD is given by:

$$\mathcal{L} = \mathcal{L}_{BCE} + \mathcal{L}_{MCE} + \lambda\mathcal{L}_{CTS}. \qquad (11)$$

### 3.6. Inference

ExVAD contains three branches that enable it to handle fine-grained and coarse-grained WSVAD tasks and anomaly interpretation. Regarding fine-grained WSVAD, we follow previous work (Wu et al., 2023) and use a thresholding strategy on the alignment graph M to predict anomalous events. For coarse-grained WSVAD, we follow previous work (Wu et al., 2024c) in employing two methods to compute the frame-level anomaly degree. The first method directly uses the anomaly scores from the coarse-grained classification, while the second method uses the alignment map from the fine-grained classification, i.e., the similarity between the video and the normal class minus 1 is the anomaly degree. Finally, we choose the best of these two methods to compute the frame-level anomaly degree.

## 4. Experiments

In this section, we perform experiments on the UCF-Crime (Sultani et al., 2019) and XD-Violence (Wu et al., 2020) datasets. Ex-VAD focuses on fine-grained anomaly detection and explainability. We compare it with other methods designed for fine-grained anomaly detection and explore novel approaches for explainable coarse-grained anomaly detection using LLMs and VLMs. Furthermore, we conduct

comprehensive ablation studies to validate the effectiveness of each module in the proposed model.

### 4.1. Experimental Setups

**Datasets.** We perform experiments on the UCF-Crime and XD-Violence datasets. **UCF-Crime** consists of 1,900 untrimmed surveillance videos with a total duration of 128 hours, covering 13 real-world anomalies (e.g., abuse, robbery, explosion) and normal activities. In the WSVAD, 1,610 videos are used for training with video-level annotations, while 290 videos are used for testing with frame-level annotations. **XD-Violence** contains 4,754 untrimmed videos totaling 217 hours, making it one of the largest multimodal violence detection datasets. It includes six types of violence (e.g., abuse, car accidents, explosions) across diverse sources such as surveillance, films, and games. The dataset is divided into 3,954 training videos and 800 testing videos, with video-level labels.

**Evaluation Metrics.** For coarse-grained WSVAD, the evaluation uses frame-level Average Precision (AP) and frame-level AUC for XD-Violence, and only frame-level AUC for UCF-Crime. For fine-grained WSVAD, mean Average Precision (mAP) values are calculated under different Intersection over Union (IoU) thresholds (ranging from 0.1 to 0.5 with a stride of 0.1). The average mAP (AVG) is also reported, and mAP is computed only for abnormal videos in the test set.

**Implementation Details.** All experiments are conducted on a single NVIDIA RTX A100 GPU using PyTorch. The network employs frozen image and text encoders from pre-trained CLIP (ViT-B/16) with a Transformer-based FFN layer and GELU activation. BLIP-2 is used for caption generation, while Llama-3.1 generates anomaly explanations. Visual and text features are fused via concatenation. Key hyperparameters include: $\sigma = 1$, $\tau = 0.07$, context length $l = 20$, window length in LGT-Adapter (64 for XD-Violence, 8 for UCF-Crime), and $\lambda$ ($1 \times 10^{-4}$ for XD-Violence, 1 for UCF-Crime).

| Category | Method | Fine-grained | Explainability | XD-Violence(AP) | UCF-Crime(AUC) |
|---|---|---|---|---|---|
| Training-Free | LAVAD(Zanella et al., 2024) | × | ✓ | 62.01 | 80.28 |
| | VERA(Ye et al., 2024) | × | ✓ | **88.2** | 86.6 |
| Fine-tuning LLMs | VADOr(Lv & Sun, 2024) | × | ✓ | - | 88.1 |
| Weakly | RealAD(Sultani et al., 2019) | ✓ | × | 75.18 | 84.14 |
| | RTFM (Tian et al., 2021b) | ✓ | × | 78.27 | 85.66 |
| | AVVD(Zhou et al., 2022b) | ✓ | × | 78.10 | 84.57 |
| | TEVAD(Chen et al., 2023) | × | × | 79.80 | 84.9 |
| | DMU(Zhou et al., 2023) | ✓ | × | 82.41 | 86.75 |
| | CLIP-TSA(Joo et al., 2023) | ✓ | × | 82.17 | 87.58 |
| | UMIL(Sánchez-Macián et al., 2024) | ✓ | × | - | 86.75 |
| | VADCLIP(Wu et al., 2024c) | ✓ | × | 84.51 | 88.02 |
| | STPrompt(Wu et al., 2024b) | ✓ | × | 83.97 | 88.08 |
| | **Ex-VAD (Ours)** | ✓ | ✓ | 86.52 | **88.29** |

*Table 3.* Coarse-grained comparisons of methods on XD-Violence and UCF-Crime datasets.

| Visual | AEGM | | LAAM | AUC (%) |
|---|---|---|---|---|
| | Captionl | Explainable text | | |
| ✓ | | | | 86.76 |
| ✓ | ✓ | | | 86.33 |
| ✓ | ✓ | ✓ | | 87.86 |
| ✓ | ✓ | ✓ | ✓ | **88.29** |

*Table 4.* Effectiveness of each module for Coarse-grained anomaly detection.

| AEGM | 0.1 | 0.2 | 0.3 | 0.4 | 0.5 | AVG |
|---|---|---|---|---|---|---|
| Captions | **17.74** | **13.27** | **10.25** | **7.01** | **6.10** | **10.88** |
| Explainable Text | 16.51 | 12.35 | 9.41 | 7.82 | 4.65 | 10.15 |

*Table 5.* Effectiveness of the Anomaly Explainable Generation Module for fine-grained anomaly detection.

| LAAM | 0.1 | 0.2 | 0.3 | 0.4 | 0.5 | AVG |
|---|---|---|---|---|---|---|
| [CLS] | 14.38 | 10.54 | 6.92 | 5.03 | 2.51 | 7.87 |
| a video of [CLS] | 14.77 | 10.68 | 6.69 | 4.78 | 3.73 | 8.13 |
| Learnable-Prompt | 15.18 | 12.03 | 6.65 | 4.96 | 3.20 | 8.40 |
| Label-Augment Prompt | **16.51** | **12.35** | **9.41** | **7.82** | **4.65** | **10.15** |

*Table 6.* Effectiveness of the Label Augment Alignment Module for fine-grained anomaly detection.

### 4.2. Comparison Results

**Fine-grained WSVAD Results.** The fine-grained detection task is more challenging as it involves detecting the presence of anomalous events while also accurately identifying their specific categories. To demonstrate the superiority of our proposed Ex-VAD, we conduct comparisons with several VAD methods, including RealAD (Sultani et al., 2019), RTFM (Tian et al., 2021b), AVVD (Zhou et al., 2022b), DMU (Zhou et al., 2023), CLIP-TSA (Joo et al., 2023), UMIL (Sánchez-Macián et al., 2024), VADCLIP (Wu et al., 2024c), and STPrompt (Wu et al., 2024b). For fairness, CLIP (ViT-B/16) is used for all feature extractors.

Tables 1 and 2 present the fine-grained detection results on UCF-Crime and XD-Violence datasets, evaluated using mean average precision (mAP) and average accuracy (AVG) across IOU thresholds (0.1–0.5). Our Ex-VAD consistently achieves the best results, highlighting its superior performance. Specifically, Ex-VAD achieves an AVG of 9.00

on UCF-Crime, outperforming VADCLIP, STPrompt, and TCVADS by 1.32, 1.52, and 7.24, respectively. On XD-Violence, Ex-VAD achieves 28.23 AVG, exceeding these methods by 3.53, 4.79, and 11.28, respectively. Unlike methods like VADCLIP, STPrompt, and TCVADS which align visual features with text embeddings from CLIP or LLMs, Ex-VAD introduces a novel approach. Using AEGM, it prompts VLMs and LLMs to generate textual information, fuses this with visual features, and aligns the representation with labels. This generated textual information enriches semantics and enhances detection performance. Additionally, LAAM expands label semantics by converting single labels (e.g., "Abuse") into descriptive phrases (e.g., "Someone is being mistreated"), better aligning with visual-text features.

**Coarse-grained WSVAD Results.** Additionally, we compare the results of the state-of-the-art methods for coarse-grained anomaly detection, including the training-free methods LAVAD (Zanella et al., 2024) and VERA (Ye et al., 2024); fine-tuned models to achieve interpretable VADor (Lv & Sun, 2024), and the above for fine-grained anomaly detection methods.

Table 3 shows that while LAVAD and VERA are simple and interpretable due to their lack of training, they do not support fine-grained detection. Our method, Ex-VAD, performs best on the UCF dataset and second best on the XD dataset. VADOr achieves explainability through fine-tuning but lacks fine-grained detection support. For methods supporting fine-

grained detection, older approaches like RealAD underperform (75.18 AP on XD-Violence), while recent methods, including AVVD, DMU, and STPrompt, show consistent improvement. VADCLIP and TCVADS push the state of the art, with TCVADS achieving 85.58 AP on XD-Violence and 88.58 AUC on UCF-Crime. Ex-VAD uniquely combines fine-grained detection and interpretability, excelling in both. Although its performance on UCF-Crime (88.29 AUC) is marginally below TCVADS (88.58), it leads to XD-Violence, highlighting its versatility. This dual capability makes Ex-VAD an optimal choice for practical applications requiring precision and insights into detection results.

### 4.3. Model Analysis

**Ablation Study.** To evaluate the impact of the two key components, AEGM and LAAM, we conducted ablation experiments on the UCF-Crime dataset by removing one or both components from Ex-VAD, with results summarized in Table 4. The findings reveal that generating captions solely for videos degrades performance, whereas cleaning these captions and generating anomaly explanations significantly enhances it. This highlights the negative impact of low-quality captions, which often contain redundant or erroneous information, and the complementary role of high-quality anomaly explanations in improving visual performance. While AEGM is primarily designed for fine-grained anomaly detection, it also contributes to coarse-grained detection improvements.

**Effectiveness of the AEGM.** We evaluate the effectiveness of fine-grained anomaly detection for AEGM, with results shown in Table 5. The analysis shows that Captions alone outperform Explainable Text in fine-grained anomaly detection, as Captions provide frame-level semantic details, while Explainable Text offers a concise video-level summary. However, Explainable Text enhances fine-grained anomaly detection while also providing transparent, summarized explanations of anomalies at the video level. Therefore, we choose Explainable Text for the final model to balance performance and interpretability.

**Effectiveness of LAAM.** We evaluate the effectiveness of LAAM in fine-grained VAD, as summarized in Table 6. The results demonstrate that LLAM-augmented labels significantly enhance detection accuracy compared to manually defined cue words and learnable prompt-based approaches. This improvement highlights the value of leveraging LLAM to generate semantically rich and contextually relevant labels that align more effectively with the visual and textual features used for fine-grained anomaly detection.

**Effectiveness of Top-k.** Figure 4 presents the impact of different top-k values in the LAAM module on coarse-grained and fine-grained detection results, respectively. The trend graphs reveal that selecting the top 4 phrases ($k = 4$) with

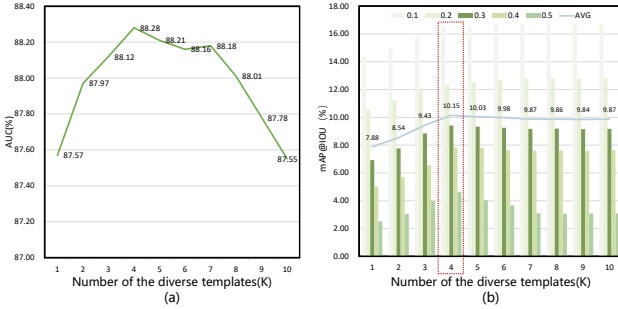

*Figure 4.* Sensitivity analysis of a different number of templates K to generalization of (a) coarse-grained detection and (b) fine-grained detection.

| Method | Trainable Params | Inference Time | MACs |
|---|---|---|---|
| RTFM | 24.72M | **8.28ms** | 126.59G |
| DMU | **6.49M** | 16.60ms | 21.00G |
| CLIP-TSA | 16.41M | 18.33ms | 102.63G |
| VADCLIP | 35.17M | 22.30ms | 29.17G |
| ExVAD | 9.97M | 15.37ms | **12.04G** |

*Table 7.* Comparison of Trainable Parameters, Inference Time, and Multiply-Add Operations (MACs). The best and second-best values are highlighted with bold text and underlined text, respectively.

the highest similarity to the original labels achieves optimal label enhancement for video anomaly detection. In this setting, the AUC for coarse-grained detection peaks at 88.28%, while the average mAP@IOU for fine-grained detection reaches its highest value of 10.15%, demonstrating the best detection performance. However, excessive enhancement ($k > 5$) may introduce noise, resulting in performance degradation. These results highlight that moderate label enhancement significantly enhances the model's overall detection capability and anomaly localization accuracy.

**Analysis of Computational Efficiency.** We evaluate the number of trainable parameters (Trainable Params), inference time of a frame (Inference Time), and multiply-add operations (MACs). Table 7 demonstrates that our method, ExVAD, achieves a well-balanced trade-off between model complexity and size, optimizing both performance and resource usage effectively.

## 5. Conclusion

In this paper, we propose Ex-VAD, an explainable approach for fine-grained video anomaly detection. First, the Anomaly Explanation Generation Module (AEGM) extracts and refines frame-level captions using VLMs, and then generates video-level anomaly explanations with an LLM. Second, the Multimodal Anomaly Detection Module (MADM) encodes the text and extracts temporal and spatial

features to detect coarse-grained anomalies. Finally, the Label Augment and Alignment Module (LAAM) expands and aligns anomaly category labels with multimodal features to achieve fine-grained anomaly detection. Experiments show that Ex-VAD outperforms existing methods in fine- and coarse-grained anomaly detection, providing a more transparent and effective solution.

## Acknowledgements

This work was supported in part by the National Natural Science Foundation of China (No.62301621), in part by Shenzhen Science and Technology Program (No. 20231121172359002, No. KQTD20221101093559018), in part by Shenzhen General Research Project (No. JCYJ20241202125904007), in part by Guangdong Basic and Applied Basic Research Foundation (No. 2025A1515011398, No.2023B0303000010), in part by the CIE-Smartchip research fund (No.2024-08), in part by the Fundamental Research Funds for the Central Universities, Sun Yat-sen University under Grants No. 23xkjc010.

## Impact Statement

This paper presents work whose goal is to advance the field of Computer Vision. There are many potential societal consequences of our work, none of which we feel must be specifically highlighted here.

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

# A. appendix.

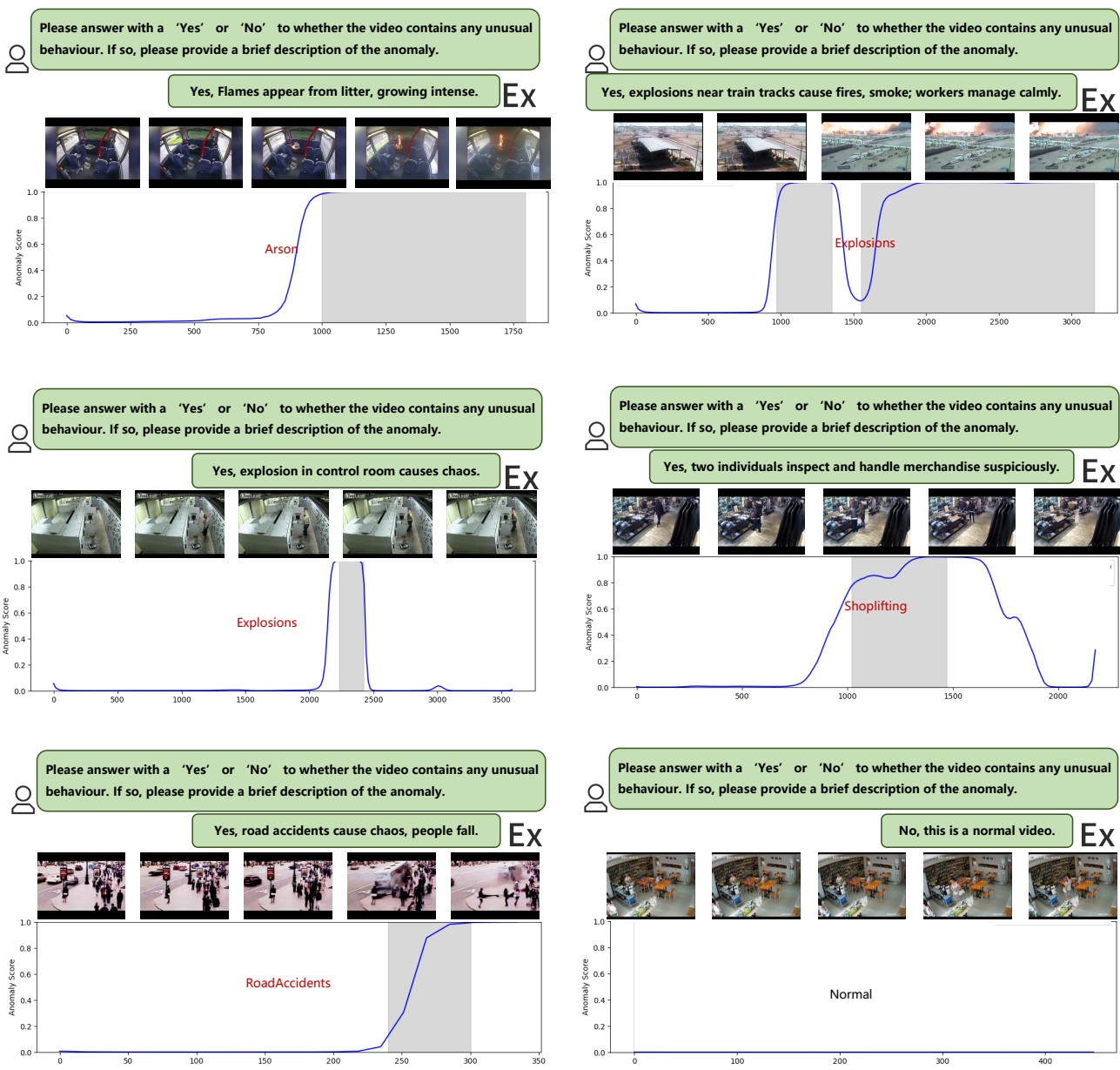

*Figure 5.* Confidence visualization on the UCF-Crime dataset.

**Qualitative Analyses.** Figure 5 illustrates the qualitative visualization of Ex-VAD. The blue curve represents the anomaly prediction score, while the grey area highlights the ground truth anomaly time positions. The figure also showcases fine-grained anomaly categories and anomaly explanations, which are generated by querying the LLM. As shown, Ex-VAD effectively detects unused anomaly categories, describes anomalous phenomena, and accurately differentiates between normal and abnormal clips in anomalous videos.

