# OpenReview forum: "Ex-VAD: Explainable Fine-grained Video Anomaly Detection Based on Visual-Language Models"
_ICML.cc/2025/Conference — ICML 2025 poster_

### Official Review · Reviewer_VnC3 · 2025-03-09

**Overall Recommendation:** 2

**Summary:**

This paper proposed an explainable approach named Ex-VAD for fine-grained video anomaly detection, which consists of three modules, Anomaly Explanation Generation Module (AEGM), Multimodal Anomaly Detection Module (MADM), and Label Augment and Alignment Module (LAAM). AEGM tries to extract and refine frame-level captions using VLMs, and then generates video-level anomaly explanations with an LLM. MADM encodes the text and extracts temporal and spatial features to detect coarse-grained anomalies. LAAM expands and aligns anomaly category labels with multimodal features to achieve fine-grained anomaly detection. Experimental results verify the effectiveness of the proposed method.

**Claims And Evidence:**

The claims made in the submission are supported by clear and convincing evidence

**Essential References Not Discussed:**

[1] Holmes-vad: Towards unbiased and explainable video anomaly detection via multi-modal llm. ArXiv2024.

**Ethical Review Flag:**

Flag this paper for an ethics review.

**Experimental Designs Or Analyses:**

The designs of Anomaly Explanation Generation Module (AEGM), Multimodal Anomaly Detection Module (MADM), and Label Augment and Alignment Module (LAAM) are reasonable.

**Methods And Evaluation Criteria:**

Yes

**Other Comments Or Suggestions:**

Please refer to the Weaknesses for more details.

**Other Strengths And Weaknesses:**

Strengths:

1.	An intuitive and motivative method is proposed for fine-grained video anomaly detection and can conduct video-level anomaly explanations.

2.	This paper is well written and easy to follow.

3.	Experimental results verify the effectiveness of the proposed method.

Weaknesses:

1.	This paper incorporates multiple existing techniques, such as VLM and LLM, to generate explainable text and uses them to improve fine-grained video anomaly detection performance. However, this is an intuitive method, and more model designs or insightful ideas should be included. Additionally, the long pipeline will affect the inference speed. Authors should discuss the inference speed.

2.	Multiple large and powerful models such as BLIP-2 are used, which makes the comparison with existing methods unfair. Additionally, why can’t use the intermediate features of BLIP-2 to improve the performance? For example, we can concatenate the intermediate features of BLIP-2 and visual features of videos or treat the intermediate features of BLIP-2 as visual features.

3.	Explainable texts are not showed in the main text. Some cases should be showed to indicate the behaviors of the proposed method.

4.	Existing explainable video anomaly detection methods such as [1] are not discussed and compared.

[1] Holmes-vad: Towards unbiased and explainable video anomaly detection via multi-modal llm. ArXiv2024.

**Questions For Authors:**

Please refer to the Weaknesses for more details.

**Relation To Broader Scientific Literature:**

results

**Theoretical Claims:**

No theoretical claims included in the paper

---

> ### Author Rebuttal · Authors · 2025-04-01
>
> We sincerely thank the reviewers for their valuable comments. We will add these valuable comments to the revised manuscript.
>
> **R1: Evaluation of Inference Time.** Thanks for your suggestion. We would like to point out that Table 7 in our main paper provides a comparison of relevant computational metrics, including Trainable Parameters, Inference Time, and Multiply-Add Operations (MACs). Our approach achieves similar inference speeds as previous methods while improving interpretability and fine-grained anomaly detection through VLM and LLM.
>
> |Method|Trainable Params|Inference Time|MACs|UCF-Crime(AUC%)|
> | :- | :-: | :-: | :-: | :-: |
> |RTFM|24\.72M|**8.28ms**|126\.59G|85\.66|
> |DMU|**6.49M**|16\.60ms|_21\.00G_|86\.75|
> |CLIP-TSA|16\.41M|18\.33ms|102\.63G|87\.58|
> |VADCLIP|35\.17M|22\.30ms|29\.17G|_88\.02_|
> |ExVAD|_9\.97M_|_15\.37ms_|**12.04G**|**88.29**|
>
> **R2: Reasons to choose BLIP2.** The use of VLMs and LLMs is an essential part of the solution to the lack of anomaly classification and anomaly explanations in existing studies. To reduce its required computational cost and resources, we tried various ways to generate captions during the experiment, such as BILP-2, LLAMA3.2, cogVLM, etc. We found that the optimal choice that combines both efficiency and performance is BLIP-2, which generates video captions with lightweight deployment and less time.
>
> Regarding BLIP-2's intermediate features, we tried to integrate them into the model, e.g. by splicing them with video visual features. Experiments show that this method has an accuracy of 88.09% on the UCF-Crime dataset, while the anomaly explanation generated by LLM can achieve 88.29%. Therefore, we finally chose the explanation text generated by LLM which could describe the cause of the anomaly in addition to improving the accuracy by 0.2%.
>
> **R3: Explainable texts are not shown in the main text.** Because of the 8-page limit, necessary experiments (such as accuracy of fine-grained and coarse-grained anomaly detection, ablation experiments, parameter selection, model complexity, and inference time) have taken up a major part of the paper, so we have placed the explainable text in the appendix section. An example showing the logic and results of anomaly text generation is given in Figure 3 in the main text. Additionally, we'll add more explainable text in camera-ready paper.
>
> **R4:** **More methods to compare.** Thanks for your suggestion. We have compared our method with existing explainable video anomaly detection methods such as LAVAD[1], VERA[2], and VADor[3], and we will compare and discuss the suggested method Holmes-vad[4] in the camera-ready paper.
>
> Different from Homles-vad which focuses on the coarse-grained anomaly detection task (determining whether a video frame is abnormal), our method can achieve coarse-grained anomaly detection and fine-grained anomaly classification (identify the abnormal category).
>
> As Homles-vad requires more supervised information, additional introduction of WSVAD [5] to generate pseudo-labels as well as fine-tuning LLM, its coarse-grained anomaly detection results outperform our method. Three main reasons are as follows:
>
> 1. Holmes-vad incorporates more supervisory information, performs single-frame temporal annotation in constructing the dataset, and generates interpretable textual descriptions based on it, whereas our method stays consistent with the traditional video anomaly detection methods without any additional annotation.
> 1. Holmes-vad is based on the existing WSVAD [5] method which first performs prediction to get the pseudo-labels and then selects the frames with anomaly scores higher than the threshold, which focuses on the keyframes as compared to uniform sampling.
> 1. Holmes-vad needs to fine-tune the multimodal large language model and requires more computational resources for training. Experiments demonstrate that our method has only 9.97M training parameters and 15.37ms inference speed. Our method has a better balance between computational resources and effectiveness.
>
> [1] LAVAD: Harnessing large language models for training-free video anomaly detection. CVPR2024
>
> [2] Vera: Explainable video anomaly detection via verbalized learning of vision-language models. ArXiv2024.
>
> [3]VADor: Video anomaly detection and explanation via large language models. ArXiv2024.
>
> [4] Holmes-vad: Towards unbiased and explainable video anomaly detection via multi-modal llm. ArXiv2024.
>
> [5] UR-DMU: Dual memory units with uncertainty regulation for weakly supervised video anomaly detection. AAAI 2023.

---

> > ### Comment · Reviewer_VnC3 · 2025-04-02
> >
> > Thanks the authors for the response. After reading the rebuttal, I still have concerns about the inference time. This paper incorporates multiple existing techniques, such as VLM and LLM. But the latency reported in Table 7 is only 15.37 ms and the MACs is 12.04G, which is unconvincing. Since large VLM and LLM are really time-consuming, and can the authors provide detialed inference costs of each component?  In addition, in Table 4, incorporating captions to visual features can't improve the performance (86.76 VS. 86.33), can the authors give some deep explainations about this phenomenon？

---

> > > ### Author Response · Authors · 2025-04-02
> > >
> > > **R1:Inference Time.** Thanks for your suggestions. We need to clarify how the results in Table 7 are tested. We firstly pre-extract the caption with VLM and then generate explainable text by LLM. During the experimental process, training and inference are performed by loading the pre-processed data. Existing methods for testing inference time are also tested by loading pre-processed data, so the inference time in Table 7 refers to the inference time for the model to perform anomaly detection, excluding the time for extracting captions and generating the explainable text. For a complete display of the model's inference time, the following table shows the detailed inference costs of each component.
> > > |Component|Caption|Explainable Text|Anomaly Detection|Total|
> > > | :- | :-: | :-: | :-: | :-: |
> > > |Inference Time|0.0003ms|3.84ms|15.37ms|19.21ms|
> > >
> > > To process one video frame, the inference costs of extracting caption, generating explainable text, and performing anomaly detection are 0.0003ms, 3.84ms, and 15.37ms, respectively. The total inference cost of our model is 19.21ms. **In other words, the inference speed of our model is 52.06 FPS, indicating that it can meet the requirements of the practical application of real-time inference.** To reduce the inference cost caused by VLM and LLM, we adopt the following strategies: 1) we extract one caption from multiple frames due to the high similarity between neighboring frames; 2) we use video-level explainable text (i.e., one video generates one explainable text); 3) the inference time is obtained by calculating the average time to process each frame.
> > >
> > > **R2: Incorporating captions into visual features can't improve the performance.** During our experiments, we found that the accuracy decreased after extracting each frame of caption and simply fusing it with visuals. The reason is that **unprocessed frame-level captions have a large amount of noise, incomplete caption content, lack of description of global context and scene dynamics, and inability to accurately respond to visual content.** To capture the global context and dynamic description of the scene, these captions were further cleaned via LLM to remove incomplete captions and generate explainable text.

---

### Official Review · Reviewer_A7T4 · 2025-03-10

**Overall Recommendation:** 4

**Summary:**

This paper introduces Ex-VAD, an explainable fine-grained video anomaly detection method based on visual-language models and large language models. By integrating modules for anomaly explanation generation, multi-modal feature fusion, and label augmentation and alignment, Ex-VAD achieves both fine-grained classification and explainability. Experimental results show that Ex-VAD significantly outperforms existing methods on the UCF-Crime and XD-Violence datasets, demonstrating its potential in the field of video anomaly detection.

**Claims And Evidence:**

The authors claim that Ex-VAD not only detects abnormal behaviors but also generates detailed anomaly explanations, which is valuable for applications requiring precise decision-making. The experimental results support this claim, especially in the fine-grained detection task, where Ex-VAD achieves significantly higher mAP on both datasets compared to other methods. However, it is suggested that the authors further quantify the quality of the explanations to more comprehensively demonstrate their interpretability advantages.

**Essential References Not Discussed:**

While the paper does an adequate job of citing relevant literature, there may be additional recent works that are essential for a comprehensive understanding of the context of the contributions. For instance, any recent studies on the integration of visual and textual information in AI systems or advancements in explainable AI could provide further context. It would be beneficial if the authors could include these in their discussion to give a more complete picture of the research landscape.

**Experimental Designs Or Analyses:**

I have reviewed the experimental designs and analyses proposed in the paper. The methodology is sound, with appropriate use of the UCF-Crime and XD-Violence datasets, which are relevant and widely recognized within the field. However, I suggest that the authors consider adding additional experiments to further validate the robustness of Ex-VAD under various conditions.

**Methods And Evaluation Criteria:**

The proposed Ex-VAD method and its evaluation criteria are well-suited for the problem of video anomaly detection. The use of mAP (mean Average Precision) and AUC (Area Under the Curve) as primary metrics aligns with established practices in the field, ensuring that the method is assessed using standard and widely-accepted benchmarks. However, it would be beneficial if the authors could provide a more detailed discussion on the validation of the explainability of the generated text.

**Other Comments Or Suggestions:**

1. It would be helpful if the authors could provide a more detailed discussion on the limitations of Ex-VAD, including scenarios where the model might not perform optimally.
2. I recommend including a more comprehensive comparison with other recent state-of-the-art methods in video anomaly detection. This would provide a clearer picture of Ex-VAD's advantages and potential areas for improvement.

**Other Strengths And Weaknesses:**

### Strengths
1. Ex-VAD not only detects abnormal behaviors but also generates detailed explanations, which is valuable for applications requiring precise decision-making, such as surveillance. This integration of fine-grained classification and explainability is an innovation in the field.
2. The experimental results on two benchmark datasets demonstrate Ex-VAD's superior performance in both fine-grained and coarse-grained detection tasks, proving its competitiveness in video anomaly detection.
### Weaknesses
1. The LAAM module enhances detection performance by expanding label semantics, but this approach may be sensitive to the initial quality of the labels. If the labels are not accurately defined or descriptive enough, performance may degrade. The authors could explore how to maintain robustness when label quality is poor.
2. While Ex-VAD generates anomaly explanations, the paper lacks quantitative assessment of the quality of these explanations. For example, user studies or comparisons with existing explainable methods could be introduced to more comprehensively demonstrate its explainability advantages.

**Questions For Authors:**

1. Could you elaborate on how the quality and usefulness of the generated explanations were quantitatively assessed?
2. Are there any plans to optimize the computational efficiency of Ex-VAD for real-time applications, and if so, what are the expected improvements in terms of speed and resource usage?"
3. What are the next steps in your research after this study? Are there any plans to address the limitations you've identified or to expand the capabilities of Ex-VAD in new directions?

**Relation To Broader Scientific Literature:**

The key contributions of this paper are closely related to the broader scientific literature on fine-grained video anomaly detection and anomaly explanation. The paper builds upon previous work by enhancing the capability of fine-grained anomaly detection while also leveraging large language models to generate explanations for anomalies, which is an area of growing research interest. It would be beneficial if the authors could provide a more detailed discussion on how Ex-VAD compares and contrasts with other recent advancements in the field.

**Theoretical Claims:**

I have reviewed the theoretical claims presented in the paper. The paper demonstrates the effectiveness of Ex-VAD in the direction of fine-grained anomaly detection through experimental results.

---

> ### Author Rebuttal · Authors · 2025-04-01
>
> We sincerely thank the reviewers for their valuable comments. We will add these valuable comments to the revised manuscript.
>
> **R1: Robustness of the label.** We use the SOTA large language model GPT to generate M phrases for label expansion and select the top-k among them as the final labels. To ensure the generated labels remain relatively stable, k is set to be much smaller than M. Experiments demonstrate that the best performance is achieved when k=4. This approach ensures a certain level of robustness in the quality of the generated labels.
>
> **R2: Analysis of Anomaly Explanation.** Currently, evaluation methods for video-text tasks are more common in video understanding. Typical quantitative analyses compare the similarity between generated text and reference text using metrics such as BLEU, CIDEr, and METEOR. Additionally, qualitative analysis is conducted by comparing texts generated by different models, where higher relevance to the video indicates better quality. In the appendix, we provide a visualization analysis of our model, which includes interpretable text that, to some extent, reflects the model's understanding of the video. In the final manuscript, we will further supplement this aspect.
>
> **R3: More comprehensive comparison with other recent SOTA methods.** You can see the reply to Reviewer VnC3's “**R4: More methods to compare.**” for a more detailed answer.
>
> **R4: Quantitative analysis of explainable texts.** Quantitative analysis of explainable texts is our next step. Our next plan is to further standardize the process of anomalous explanatory text generation, and its qualitative analysis. Specifically, 1. optimize the text generation strategy: make the generated anomaly explanations more precise and readable by adjusting the prompt engineering or introducing more reasonable templates. 2. enhance the standardization of the qualitative analysis: introduce the qualitative assessment criteria of the text, such as BLEU, CIDEr, METEOR, etc.

---

> > ### Comment · Reviewer_A7T4 · 2025-04-07
> >
> > Thanks for the authors' reply. My concerns have been addressed, and I will maintain my current score.

---

### Official Review · Reviewer_QRsp · 2025-03-11

**Overall Recommendation:** 4

**Summary:**

This paper proposes Ex-VAD, an explainable fine-grained video anomaly detection method that integrates visual-language models (VLMs) and large language models (LLMs). The approach consists of three main modules: the Anomaly Explanation Generation Module (AEGM), the Multi-modal Anomaly Detection Module (MADM), and the Label Augment and Alignment Module (LAAM). The AEGM uses VLMs to generate frame-level captions and leverages LLMs to produce video-level anomaly explanations. The MADM combines text and visual features for coarse-grained anomaly detection, while the LAAM enhances fine-grained classification by expanding and aligning labels. Experimental results demonstrate that Ex-VAD significantly outperforms existing methods on the UCF-Crime and XD-Violence datasets. ## update after rebuttal

**Claims And Evidence:**

The authors claim that Ex-VAD outperforms existing methods in both fine-grained and coarse-grained video anomaly detection tasks. The experimental results support this claim, especially in the fine-grained detection task, where Ex-VAD achieves significantly higher mean Average Precision (mAP) compared to other methods on both datasets.

**Essential References Not Discussed:**

In addition to the weakly supervised methods and training-free methods mentioned, the authors have not taken into account some recently published approaches, such as those based on LLMs. Discussing these methods would help to clarify whether the explainability of Ex-VAD is effective.

**Experimental Designs Or Analyses:**

The experimental design is reasonable, as it compares both coarse-grained and fine-grained metrics. Although the coarse-grained metric on the UCF dataset is only 0.21% higher than the SOTA method, there is a significant improvement in the fine-grained metric, such as a 3.67% increase on the UCF dataset. These results demonstrate the effectiveness of the proposed method.

**Methods And Evaluation Criteria:**

The design of Ex-VAD is reasonable, integrating multi-modal feature fusion and label augmentation alignment, making it suitable for video anomaly detection tasks. The primary evaluation metrics, including mAP (mean Average Precision) and AUC (Area Under Curve), are standard practices in the field.

**Other Comments Or Suggestions:**

If possible, please include comparisons with some of the latest methods from 2025.

**Other Strengths And Weaknesses:**

Strengths:
+ Ex-VAD combines fine-grained classification with anomaly explanations, filling the gap in explainability of existing video anomaly detection methods. This is particularly important for applications requiring precise responses, such as surveillance and healthcare monitoring.
+ By integrating text information generated by the AEGM with visual features, Ex-VAD leverages the strengths of both modalities to significantly enhance anomaly detection performance.
+ The extensive experiments on two benchmark datasets show that Ex-VAD outperforms existing methods in both fine-grained and coarse-grained detection tasks, demonstrating its effectiveness and superiority.
Weaknesses:
+ Although Ex-VAD performs well in terms of computational efficiency, its reliance on multiple complex modules (e.g., VLMs and LLMs) may lead to higher resource consumption in practical deployment. The authors could further explore model lightweighting or optimization strategies.
+ Ex-VAD relies heavily on pre-trained VLMs and LLMs, which may limit its applicability to specific domains or custom datasets. You may investigate how to fine-tune the model with limited data to improve its performance in specific tasks.

**Questions For Authors:**

+ Could you elaborate on how you specifically selected and optimized the strategy for multi-modal feature fusion during the training process?
+ Are there plans to extend the Ex-VAD method to other types of datasets or domains in future work, such as anomaly detection and early warning in autonomous driving?

**Relation To Broader Scientific Literature:**

This method is closely related to the previous VADCLIP approach but differs in that it utilizes LLMs to generate anomaly explanations for videos as textual information rather than relying solely on visual information. Compared to previous methods, this textual information adds more detail to the videos, complementing the visual information and significantly enhancing the capability for fine-grained anomaly detection.

**Theoretical Claims:**

I have carefully reviewed the theoretical claims made in the paper. The authors argue that introducing the text modality can further enhance the capability of anomaly detection, especially for fine-grained anomaly detection. This reasoning is sound and aligns well with the goals of the paper, and the experimental results also validate this conclusion.

---

> ### Author Rebuttal · Authors · 2025-04-01
>
> We sincerely thank the reviewers for their valuable comments. We will add these valuable comments to the revised manuscript.
>
> **R1: Resource consumption.** We appreciate your concern regarding the resource consumption of Ex-VAD due to the integration of VLMs and LLMs. We also recognize that due to the integration of multiple complex modules such as VLM and LLM, it may impose high resource requirements when deployed. Therefore, in the experimental session, we compared the training parameters and Multiply-Add Operations(MACs). Table 7 demonstrates that our method achieves a well-balanced trade-off between model complexity and size, optimizing both performance and resource usage effectively.
>
> |Method|Trainable Params|Inference Time|MACs|UCF-Crime(AUC%)|
> | :- | :-: | :-: | :-: | :-: |
> |RTFM|24\.72M|**8.28ms**|126\.59G|_85\.66_|
> |DMU|**6.49M**|16\.60ms|_21\.00G_|86\.75|
> |CLIP-TSA|16\.41M|18\.33ms|102\.63G|87\.58|
> |VADCLIP|35\.17M|22\.30ms|29\.17G|88\.02|
> |ExVAD|_9\.97M_|_15\.37ms_|**12.04G**|**88.29**|
>
> **R2: Fine-tune the model.** We appreciate your concern and suggestion about fine-tuning the model. At the beginning of our experiments, we thought that there were two ways to obtain anomaly explanations: the first way is to fine-tune the LLM to achieve the interpretability of anomaly videos, and the second way is to invoke the knowledge of the LLM by setting the cue words to obtain the desired anomaly explanations. Through the experiments, we found that the computational resources, model complexity, and reasoning speed required for fine-tuning LLM are much higher than that of the method of setting cue words when there is not much difference in accuracy. Therefore, ultimately, our approach is to efficiently utilize the general knowledge of LLM by setting cue words to achieve anomaly interpretation and improve detection accuracy.
>
> **R3: Multi-modal feature fusion during the training process.** During the training process, we optimize and select the multimodal fusion strategy in the following two steps. Firstly, we refer to the fusion strategies commonly used by previous studies, such as contact, cross-modal attention, and addition, etc. Secondly, we try the above methods in the experimental process and select the optimal fusion method by comparing the accuracy.
>
> **R4: Future work.** Thanks for your attention, our next plan is to further standardize the process of anomalous explanatory text generation, and its qualitative analysis. Specifically, 1. optimize the text generation strategy: make the generated anomaly explanations more precise and readable by adjusting the prompt engineering or introducing more reasonable templates. 2. enhance the standardization of the qualitative analysis: introduce the qualitative assessment criteria of the text, such as BLEU, CIDEr, METEOR, etc.

---

### Official Review · Reviewer_p4Ua · 2025-03-13

**Overall Recommendation:** 3

**Summary:**

Paper proposes an explainable VAD approach which combines fine-grained classification with explanations. The approaches use pre-trained VLM and LLM to extract the relevant features. The approach employed 3 linear combination of 3 loss functions for the fine-grained classification of anomalous videos. A novel label-enhanced alignments method was used to optimize the feature fusion. Experiments on 2 popular VAD datasets show promising results against the SOTA methods.

**Claims And Evidence:**

The experimental results in Table 1 (UCF-Crime), Table 2 (XD-Violence) and Table 3 (Coarse grained, UFC-Crime, XD-Violence) shows that the proposed method outperform other SOTA methods.

**Essential References Not Discussed:**

The earlier work in 2023 using VLM for VAD is not cited and discussed.

Chen, W., Ma, K. T., Yew, Z. J., Hur, M., & Khoo, D. A. A. (2023). TEVAD: Improved video anomaly detection with captions. In Proceedings of the IEEE/CVF Conference on Computer Vision and Pattern Recognition (pp. 5549-5559).

**Ethical Review Concerns:**

Not applicable.

**Experimental Designs Or Analyses:**

No major issue on experimental designs and analysis.

The exclusion of other benchmarks, like ShanghaiTech and UCSD-Ped2 should be explained to avoid suspicion of cherry-picking results.

**Methods And Evaluation Criteria:**

The methods and evaluation criteria are sensitive. However, there are other VAD datasets which should also be included in the experiments, e.g. ShanghaiTech [1] and UCSD-Ped2 [2]


[1] Dan Xu, Rui Song, Xinyu Wu, Nannan Li, Wei Feng, and Huihuan Qian. Video anomaly detection based on a hierarchical activity discovery within spatio-temporal contexts, Neurocomputing

[2] Wen Liu, Weixin Luo, Dongze Lian, and Shenghua Gao. Future frame prediction for anomaly detection–a new baseline.
In Proceedings of the IEEE conference on computer vision and pattern recognition

**Other Comments Or Suggestions:**

Typos:
Table 6: "Lable Augmnet" should be "Label Augment"

**Other Strengths And Weaknesses:**

Strengths
1. Paper's experimental results are strong for the 2 benchmarks experimental upon.
2. The use of pretrained VLM models for explainable VAD is somewhat novel. But there are some prior works.
3. Label Augment and Alignment Module is incrementally novel.

Weaknesses
1. Not much technical contributions. Main novelty is the pipeline of modules and the Label Augment and Alignment module.
2. Experimental results while strong, are not particularly significant compare to SOTA.

**Questions For Authors:**

Will the paper include the two benchmarks, ShanghaiTech and UCSD-Ped2 to show the generalizability of the proposed method?

**Relation To Broader Scientific Literature:**

Paper's approach is similar to other explanable AI using VLM. The application of fine-grained VLM is also not very novel. However, the experimental results are promising.

**Theoretical Claims:**

No theoretical claims. Equations were read but not checked.

---

> ### Author Rebuttal · Authors · 2025-04-01
>
> We sincerely thank the reviewers for their valuable comments. We will add these valuable comments to the revised manuscript.
>
> **R1：Novelty of the Proposed Pipeline.** We apologize for failing to highlight our contributions and novelty. Different from existing coarse-grained VAD, our method is unique in using the generated anomaly text for both anomaly explanation and fine-grained anomaly detection. Compared with traditional fine-grained anomaly detection, our accuracy has a large improvement, e.g., in the UCF dataset, the average mAP(AVG) is 3.67% higher compared with SOTA. Additionally, compared with fine-tuning a large language model (LLM) or a multimodal large language model (MLLM) for anomaly detection, our approach requires significantly fewer parameters (9.19M) and achieves a faster inference speed (15.37ms). This demonstrates a good trade-off between complexity and effectiveness.
>
> **R2：Reasons for not using ShanghaiTech and UCSD-Ped2.** The key idea of our method aims at the fine-grained detection and description of the causes of anomalies, and its advantage is to generate anomaly explanation text through VLMs and LLMs, which both describe the causes of video anomalies and greatly improve the accuracy of the fine-grained anomaly classification. Therefore, we select datasets that contain fine-grained anomaly detection in existing studies, such as UCF-Crime and XD-Violence, whereas the Shanghai and UCDF datasets have not been used by researchers for fine-grained anomaly detection, and thus cannot be compared with the superiority of our method.
>
> Considering your suggestion, we will add related experiments for the shanghaiTech and UCSD-Ped2 datasets. Due to time constraints, we only had time to add the results of the coarse-grained anomaly detection experiment on shanghaiTech, and will add the results on UCSD-Ped2 later. As shown in the below table, we performed coarse-grained anomaly detection on the ShanghaiTech dataset and compared it with other methods:
>
> |**Method**|Feature|AUC(%)|
> | - | - | - |
> |GAN-Anomaly(CVPR2019)|TSN|84\.4|
> |RTFM(ICCV2021)|I3D|97\.21|
> |CMRL(CVPR2023)|I3D|97\.60|
> |TEVAD(CVPR2023)|I3D|98\.10|
> |PE-MIL(CVPR2024)|I3D|**98.35**|
> |STPrompt(ACMMM2024)|CLIP|97\.81|
> |Ours|CLIP|_98\.23_|
>
> Although our method is not SOTA, it is only 0.12% less than the best result, 0.13% higher than TEVAD, and 0.42% higher than STPrompt, which also uses the clip to extract features. This experiment shows that our method is also applicable to the Shanghai dataset.
>
> **R3: Citation and discussion of TEVAD.** TEVAD makes full use of VLM to generate video captions as textual modalities and achieves improved accuracy in anomaly detection by fusing it with visual modalities. We will cite and discuss it in the final manuscript.
>
> Both TEVAD and our method utilize VLMs to generate captions, which extends the traditional single visual modality to visual and textual multimodal branching, improving anomaly detection. Our advantages over TEVAD are shown below. First, our proposed method not only determines whether a video frame is abnormal but also further identifies the abnormal category (Abuse, Arrset, etc.). Second, instead of just generating captions, the video anomalies are further leveraged to describe the video anomalies through LLMs.
>
> **R4: others.** Thank you for pointing out the spelling error, we will change Table 6: 'Lable Augmnet' to 'Label Augment' in the camera-ready paper.

---

### Decision · Program_Chairs · 2025-05-01

**Decision:**

Accept (poster)

**Comment:**

This introduces Ex-VAD, a fine-grained anomaly detection method enhanced by textual explanations from large models.

Most reviewers appreciated the integration of VLMs and LLMs for richer interpretations and better performance on UCF-Crime and XD-Violence, improved fine-grained results over previous methods.

However, there were concerns about efficiency and that some preprocessing steps may not included in the reported inference time. Also, folks want wanted broader benchmarking on datasets such as ShanghaiTech (partially addressed by the team) and UCSD-Ped2.

The authors attempted to address these points by clarifying where processing times are accounted for and adding partial experiments on ShanghaiTech. Although one reviewer remains unconvinced about practical overhead and reliability, the majority view is that the work offers a useful step toward more interpretable video anomaly detection. I am leaning to accept.